# GFlowNets with Human Feedback

**Yinchuan Li[1], Shuang Luo[1,2]\*, Yunfeng Shao[1], Jianye Hao[1,3]**
[1]Huawei Noah's Ark Lab, Beijing, China   [2]Zhejiang University, Huangzhou, China
[3]Tianjin University, Tianjin, China

## Abstract

We propose the GFlowNets with Human Feedback (GFlowHF) framework to improve the exploration ability when training AI models. For tasks where the reward is unknown, we fit the reward function through human evaluations on different trajectories. The goal of GFlowHF is to learn a policy that is strictly proportional to human ratings, instead of only focusing on human favorite ratings like RLHF. Experiments show that GFlowHF can achieve better exploration ability than RLHF.

## 1 Introduction

Large-scale language models are one of the main applications of artificial intelligence, represented by ChatGPT, which has become a hot trend (Ouyang et al., 2022; Stiennon et al., 2020; Nakano et al., 2021; Ziegler et al., 2019; Thoppilan et al., 2022). Reinforcement Learning from Human Feedback (RLHF) (Christiano et al., 2017) techniques play a key role in ChatGPT. However, RL suffers from insufficient exploration ability since it tends to take actions that maximize the expectation of future rewards. Therefore, Generative Flow Networks (GFlowNets) (Bengio et al., 2021b) have been recently proposed to make up for the insufficient exploration of RL, which generate distribution proportional to the rewards and have been used in many applications (Bengio et al., 2021a; Zhang et al., 2022; Deleu et al., 2022; Li et al., 2023a;b; 2022). This property makes GFlowNets ideal for training AI models with human feedback, helping us explore more diverse outcomes.

In this paper, we propose Generative Flow Networks with Human Feedback (GFlowHF). Our main contributions lie in proposing, to the best of our knowledge, the first GFlowNets with human feedback framework, and conducting experiments in a diverse reward distribution environment to validate that GFlowHF can obtain more diversity high-scoring answers than RLHF with the same human labels, and has a stronger ability to resist noisy labels than RLHF.

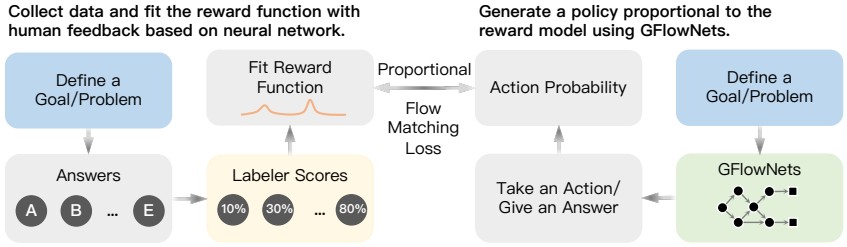

Figure 1: Overall framework of GFlowHF.

## 2 GFlowHF Framework

The overview framework of GFlowHF is shown in Figure 1. Considering a task with tuple $(\mathcal{S}, \mathcal{A})$, where $\mathcal{S}$ denotes the state space and $\mathcal{A}$ denotes the action space. Define a complete trajectory $\tau = (s_0, ..., s_f)$ as a sequence sampled states starting in $s_0$ and ending in $s_f$. During training, we sample a set of trajectories $\{\tau_1, ..., \tau_N\}$ and send them to humans for scoring. We can score based on the entire trajectory, or directly based on the final state $s_f$, denoted by $\text{score}(s_f)$. Based on manual scoring, we can train a reward network $r_\phi(s_f)$ with $s_f$ as the input and $\text{score}(s_f)$ as the output.

---

\*Corresponding Author: Shuang Luo (e-mail: luoshuang@zju.edu.cn). This work was completed while Shuang Luo was a member of the Huawei Noah's Ark Lab for advanced study.

The policy of GFlowHF is defined as $\pi(a_t|s_t) = \frac{F_\theta(s_t, a_t)}{F_\theta(s_t)}$, where $F_\theta$ is the flow network. The goal of GFlowHF is to train a flow network satisfying $\pi(s_f) \propto \text{score}(s_f)$. Once $r_\phi(s_f)$ is available, we train a flow network $F_\theta$ based on the flow matching loss

$$\mathcal{L}(\tau) = \sum_{s_t = s_1}^{s_f} \left( \sum_{s_{t-1} \in \mathcal{P}(s_t)} F_\theta(s_{t-1} \to s_t) - r_\phi(s_t) - \sum_{s_{t+1} \in \mathcal{C}(s_t)} F_\theta(s_t \to s_{t+1}) \right)^2 \quad (1)$$

for discrete space tasks, where $r_\phi(s_t) = 0$, $\mathcal{P}(s)$ and $\mathcal{C}(s)$ denote the parent set and child set of $s$, respectively.

**Why GFlowHF?** The main advantage of GFlowHF over RLHF is that it can learn the distribution of rewards. Therefore, instead of simply ranking the answers, we can train the model by scoring the answers. As shown in Figure 2 Left, the preference rankings can be inaccurate. If a labeler is asked to rate several bad answers, the highest-ranked answer is actually scored poorly, and vice versa. Hence, it is more accurate to train the model by using the scores (e.g. preference percentage). Note that it is also possible to train GFlowHF directly with the ranked labels if in some cases we do not have the preference scores for the answers.

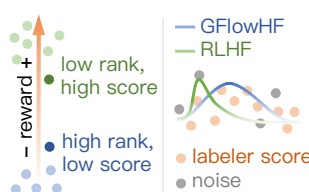

Figure 2: Advantages.

In addition, GFlowHF has a stronger ability to resist noisy labels. It is generally assumed that 10% of labels placed by humans are almost randomly distributed (Christiano et al., 2017). If a wrong label has a high score, RL focuses on learning high-scoring results and gets the wrong policy. In contrast, GFlowNet tends to learn the entire distribution and is more robust (see Figure 2 Right).

## 3 EXPERIMENTAL RESULTS

We compare the proposed GFlowHF with several RLHF methods in Point-Robot task. We set two different goals (with coordinates $(5, 10)$ and $(10, 5)$) to simulate a multimodal reward task. The agent starts at the starting coordinate $(0, 0)$ and moves towards the goals with a maximum step length of 12. After reaching the last step, we let the labeler give a preference score based on the final position. The preference score is set to be divided into 5 grades, the closer to goals, the greater the score. For the continuous Point-Robot task, our GFlowHF is developed based the CFlowNets proposed in (Li et al., 2023c). For a multimodal reward distribution, i.e. there are many different high-scoring answers. We can see that GFlowHF can demonstrate an exploration ability far beyond RLHF, i.e., it can learn all answers, while RLHF can only learn one. The number of valid-distinctive answers of GFlowHF is higher, and the reward is also slightly higher than RLHF methods.

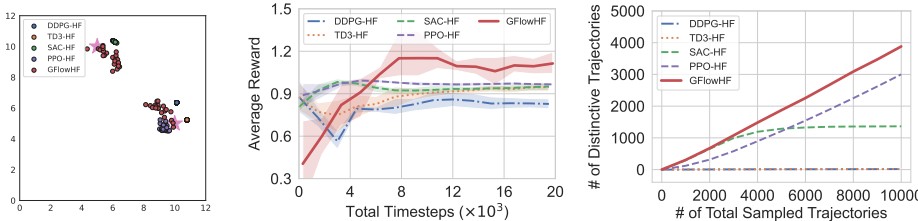

Figure 3: Comparison results of GFlowHF and several RLHF methods on Point-Robot task. **Left:** Explored answers. **Middle:** Average reward. **Right:** Number of valid-distinctive answers. GFlowHF explores both targets whereas RLHF techniques only explore one of them.

## 4 CONCLUSION & FUTURE WORK

We propose the GFlowNets with Human Feedback framework to improve the exploration when training AI models. Experiments show that GFlowHF can achieve better exploration ability and has a stronger ability to resist noisy labels than RLHF. Our future work is to use GFlowHF to train a powerful large-scale language model based on real language data.

URM STATEMENT

The authors acknowledge that the first and second authors of this work meet the URM criteria of the ICLR 2023 Tiny Papers Track. The first author is 28 years old and non-white. The second author is 25 years old and non-white student.

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

## A APPENDIX

We compare the proposed GFlowHF with several RLHF methods based DDPG (Lillicrap et al., 2015), TD3 (Fujimoto et al., 2018), PPO (Schulman et al., 2017), and SAC (Haarnoja et al., 2018). We provide the hyper-parameters of all compared methods in Table 1. The number of sample flows, action probability buffer size, and $\epsilon$ of GFlowHF are set as 100, 1000, and 1, respectively. The target update interval of SAC-HF is set as 1. The GAE parameter, timesteps per update, number of epochs, clipping parameter and value loss coefficient of PPO-HF are set as 0.95, 2048, 10, 0.2, and 0.5, respectively. A valid-distinctive answer is defined as a reward above a threshold $\delta_r$ while the MSE between the trajectory and other trajectories is greater than another threshold $\delta_{\mathrm{mse}}$.

As for the continuous Point-Robot task, assume that $K$ flows are sampled, we can use the sampled flow matching loss proposed in (Li et al., 2023c) as the following

$$\mathcal{L}_\theta(\tau) = \sum_{s_t=s_1}^{s_f} \left[ \sum_{k=1}^{K} F_\theta(G_\psi(s_t, a_k), a_k) - \lambda r_\phi(s_t) - \sum_{k=1}^{K} F_\theta(s_t, a_k) \right]^2, \qquad (2)$$

where $G_\psi$ is a pre-trained network to find the parent states and $\lambda = K/\mu(\mathcal{A})$.

Table 1: Hyper-parameters of all compared methods.

|  | GFlowHF | DDPG-HF | TD3-HF | SAC-HF | PPO-HF |
|---|---|---|---|---|---|
| Total Timesteps | 20,000 | 20,000 | 20,000 | 20,000 | 20,000 |
| Start Traning Timestep | 1,500 | 1,500 | 1,500 | 1,500 | - |
| Max Episode Length | 12 | 12 | 12 | 12 | 12 |
| Network Hidden Layers | Flow [256,256] | Actor [256,256] | Actor [256,256] | Actor [256,256] | Policy [256,256] |
| Network Hidden Layers | Retrieval [256,256,256] | Critic [256,256] | Critic [256,256] | Critic [256,256] | Value [256,256] |
| RewardNet Hidden Layers | [256,256,256] | [256,256,256] | [256,256,256] | [256,256,256] | [256,256,256] |
| Optimizer | Adam | Adam | Adam | Adam | Adam |
| Learning Rate | 0.0003 | 0.0003 | 0.0003 | 0.0003 | 0.0003 |
| Batchsize | 128 | 256 | 128 | 1024 | 64 |
| Replay Buffer Size | 8,000 | 100,000 | 100,000 | 100,000 | - |
| Discount Factor | - | 0.99 | 0.99 | 0.99 | 0.99 |
| Target Network Update Rate | - | 0.005 | 0.005 | - | - |

In addition, we compare the proposed GFlowHF with RLHF algorithms under noisy human labels. We add a noisy label at coordinate $(7, 10)$ with preference score 6. It can be seen from Figure 4 that for noisy environments, GFlowHF can still obtain the correct answers, which shows that it is very robust to noise. Although human label at $(7, 10)$ is wrong, the overall reward distribution trend will not change. GFlowHF learns the overall distribution trend and can still come up with the correct answer. In contrast, the RLHF algorithms tend to focus on where the reward is largest. If the wrong label scores high, the RLHF algorithms will learn the wrong policy, causing the RLHF algorithm to be easily disturbed by noise and give wrong answers. It is worth noting that the reward obtained by GFlowHF here is not the highest. This is because RL mostly picks wrong answers and thus gets higher rewards. This experiment demonstrates the importance of the GFlowHF algorithm in practice, especially for large language model training tasks where human label errors are common.

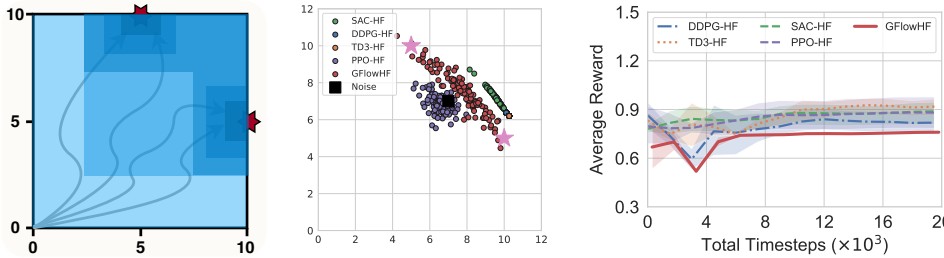

Figure 4: Comparison results of GFlowHF and several RLHF methods with noisy human labels. **Left:** Labeled score distribution. The blue color from light to dark represents the 5 grades rated by humans. **Middle:** Explored answers. **Right:** Average reward.

