# OpenReview forum: "GFlowNets with Human Feedback"
_ICLR.cc/2023/TinyPapers — Submitted to Tiny Papers @ ICLR 2023_

### Official Review · Reviewer_ALnE · 2023-03-28

**Confidence:** 3

**Summary Of Contributions:**

The paper introduce GFlowNetsHF - a framework incorporating GFlowNets instead of Reinforcement Learning (RL) to improve upon the Reinforcement Learning with Human Feedback (RLHF) approach for improving the performance of large language models.

**Rating:**

Great Start (GS): a submission which meets some of the reviewing criteria but has room for improvement

**Strengths And Weaknesses:**

Strengths:
The authors propose a new method based on GFlowNets for incorporating human feedback that can outperform RLHF because it can score answers and not just rank them like RLHF. Thus, it can learn the distribution of rewards. This method is also more robust to noisy labels which is not an uncommon occurrence in human-labeled data.

Weaknesses:
- Quite a few assumptions have been made about the background of the reader.
- The paper can definitely improve upon the clarity and completeness of the work. Many fundamentals for understanding the research problem have been glossed over.
- As for reproducibility, the experimental results section is poorly explained and the appendix does not cover all the details of the experimental setup.

**Suggested Changes:**

It would be great if the authors could explain the details of the paper better. The problem statement, motivation, background and approach must be thoroughly covered in the main paper and the appendix could contain the experimental details better.

---

### Official Review · Reviewer_u8sn · 2023-03-31

**Confidence:** 3

**Summary Of Contributions:**

The authors use GFlowNets to tackle the exploration problem in RLHF. This allows earning policies that are proportional to the score but are not focused on optimizing for the top scores since they can often be noisy. The authors evaluate the performance of GFlowNets on a point robot task with multimodal reward distributions and demonstrate superior performance against some SOTA RL baselines. They additionally demonstrate the robustness of GFlowNets against noisy labels.

**Rating:**

Clear, Correct, and Reproducible (CCR): a submission which meets the reviewing criteria

**Strengths And Weaknesses:**

Strengths:
- This is a promising research direction, and very well timed, since with the onset of LLMs RLHF is coming increasingly in the limelight
- GFlowNets can be very powerful for dsitributional reward settings and noisy preferences, and I feel exploring this direction might open up potential applications into many other multimodal scenarios
- The experimental setup has been communicated in a clear and reproducible manner

Weaknesses:
- Generally, I think the paper is well-written, but needs some adjustments in terms of clarity of thought in certain sections and grammatical errors. Please see the suggested changes for that
- To me, what comes out as a potential conclusion is that methods that work with the reward distribution can potentially perform better than RLHF methods that optimize for the best reward. This is not a new argument in the standard RL literature, since methods in Distributional RL (e.g. C51, IQNs) [1,2] have already talked about the benefits of distributions. What did not come out very clearly to me was why exactly we need GFlowNets. Potentially, this can be addressed by either showing that GFlowNets are better than these methods (i.e. adding them as baselines, or citing publications that already talk about this issue) or discussing how the setting that the authors tackle is different from what these methods address. I would recommend looking into this and making the necessary changes.

[1] [https://arxiv.org/pdf/1707.06887.pdf](https://arxiv.org/pdf/1707.06887.pdf)
[2] [https://arxiv.org/pdf/1806.06923.pdf](https://arxiv.org/pdf/1806.06923.pdf)

**Suggested Changes:**

- Grammatical mistakes. Please check the paper thoroughly for these. Here are some examples:
    - Therefore, Generative Flow Networks (GFlowNets) Bengio et al. (2021)  **is** recently proposed to make up for the insufficient exploration of RL … — should be **have been**
    - … and conducting experiments in a **diversity** reward distribution environment — should be **diverse**
- I think the environment needs a bit more explanation in the appendix
    - What is a preference, and how does it exactly work?
    - What is the scale of reward achievable — when do we consider the environment solved?

---

### Author Response · Authors · 2023-05-31
**New paper revision**

Dear ICLR Program Chairs:

We have uploaded a new revision of our paper. We have fully considered the reviewer's comments and revised accordingly.

We also hope that our paper can be presented (notable).

Best regards

---

### Comment · Area_Chair_a638 · 2023-06-06
**Final meta-review: Invite to archive**

This work meets the threshold for archival, contains the URM statement and is deanonymized

---

### Meta-Review · Area_Chair_a638 · 2023-04-05

**Recommendation:** Invite to present
**Confidence:** 4

**Metareview:**

Thank you for this exciting and promising work. This paper introduces a new framework -- GFlowNets with human feedback. The significance is that GFlowNets find policies which more evenly cover the reward distribution compared to RL methods which tend to converge on optimal solutions (and are therefore fallible to noise in human responses). Both reviewers agree this is a promising research direction and that the work is well generally well done. The main issue is that the paper is framed strongly as a paper about LLMs paper (for which it definitely has applications) whereas it is in fact primarily an RL-style paper.

Pros:
* An interesting and timely problem setup, and a well motivated solution making this paper likely to have high impact.
* Authors show their model beats some relevant SOTAs and baselines.
* Interesting use-case for LLMs pointed out.
* Nice, informative figures.

Cons:
* This work is incremental on the original GFlowNets paper.
* The link to LLMs is overstated: though, as the authors correctly point out, LLMs are an interesting application of this framework this is _not_ what they are doing here.
* It is misleading to start this paper with a paragraph about LLMs. It is mainly a paper about RL, not LLMs. This is more than just a critique in presentations style...it took me a long time to understand this paper because I thought the authors were training LLMs, which they aren't.
* It cannot be understood without referencing the task setup in the Appendix.
* Grammatical errors can make it hard to understand.
* Makes some fairly significant assumptions on the readers background.

One reviewer rates this as clear, correct and reproducible (CCR), the other as a great start (GS). Given that this paper is well done and the suggested changes from the reviewers are minor and can be solved quickly, I am willing to recommend this paper for a presentation. If the authors could weaken the LLM slant, therefore making the contribution clearer, solve the grammatical errors, fix a majority of the suggested changes from the reviewers, and clarify the task setup inside the main body I would be willing to bump up the recommendation on this potentially impactful work.

**Summary:**

This paper introduces a new framework -- GFlowNets with human feedback. The significance is that GFlowNets find policies which more evenly cover the reward distribution compared to RL methods which tend to converge on optimal solutions (and are therefore fallible to noise in human responses). Both reviewers agree this is a promising research direction but the paper is framed too strongly as a paper about LLMs, misleading the reader.

**Comments And Feedback To The Authors:**

The functional form of $\pi$ is unclear. Is it a function over $a_t$ given $s_t$ (as in the first line of page 2) or is it a scalar function over $s$ (as in the second line of page 2). To be a policy which maps states to actions does this mean the action space here is scalar and if so could you clarify this when you introduce $\mathcal{S}$ and $\mathcal{A}$.

Overall, for readers who are not familiar with this area of research or training LLMs what does it even mean to take an action? If this could be added in the remaining space the paper would be significantly more approachable for the lay-reader. This comment relates to that made by reviewer u8sn about it being important to expand on the environment structure in the appendix a bit more (I would advocate at least a sentence make it into the main body). At a minimum it is essential to add to figure 3 caption something like: ``notice how GFlowHF explores both rewards whereas RLHF technique only explore one of them''

I suspect the problem here is that this paper is worthy of not just being a `Tiny paper' but actually a full-length conference paper. The authors should feel encouraged to carry on this exciting work and I look forward to reading an extended version!

If you need more space I would recommend taking out the mathematics for continuous space tasks (discrete is enough) and put it in the appendix.

You could usefully add on to figure 4 (left) some sample trajectories which an optimised RL algorithm would find in a different colour.

More detailed figure captions would help

Typos (some also pointed out by reviewers)
* the highest-ranked answer is actually _score_ poorly --> _scored_ ?


**Reason For Not Giving A Higher Recommendation:**

I would have liked to rate this paper higher. I think it is a really nice idea, well executed with promising results. Unfortunately the presentation is slightly misleading. I do not think the authors need to slant this work so strongly towards a potential application for LLMs. A single sentence or two would suffice.

**Reason For Not Giving A Lower Recommendation:**

This work could potentially have very high impact given it is the first instance of policy learning with human feedback with GFlowNets and the potential application to LLMs.

---

### Decision · Program_Chairs · 2023-04-07

**Decision:**

Invite to present

**Comment:**

Please add your URM statement.